

# Influences of oceanic ozone deposition on tropospheric photochemistry

Ryan J. Pound[1], Tomás Sherwen[1,2], Detlev Helmig[3], Lucy J. Carpenter[1], and Mat J. Evans[1,2]

[1]Wolfson Atmospheric Chemistry Laboratories, Department of Chemistry, University of York, York, YO10 5DD, UK
[2]National Centre for Atmospheric Science, University of York, York, YO10 5DD, UK
[3]Institute of Alpine and Arctic Research, University of Colorado at Boulder, Boulder, Colorado, USA.

**Correspondence:** Ryan J. Pound (rp819@york.ac.uk)

**Abstract.** The deposition of ozone to seawater is an important ozone sink. Despite constituting as much as a third of the total ozone deposition, it receives significantly less attention than the deposition to terrestrial ecosystems. Models have typically calculated the deposition rate based on a resistance-in-series model with a uniform waterside resistance. This leads to models having an essentially uniform deposition velocity of approximately $0.05 \, \mathrm{cm \, s^{-1}}$ to seawater, which is significantly higher than the limited observational dataset. Following from Luhar et al. (2018) we include a representation of the oceanic deposition of ozone into the GEOS-Chem model of atmospheric chemistry and transport based on its reaction with sea-surface iodide. The updated scheme halves the calculated annual area-weighted mean deposition velocity to water from $0.0464 \, \mathrm{cm \, s^{-1}}$ ($25^{th}$ and $75^{th}$ percentiles of $0.0461 \, \mathrm{cm \, s^{-1}}$ and $0.0471 \, \mathrm{cm \, s^{-1}}$ respectively), to $0.0231 \, \mathrm{cm \, s^{-1}}$ ($25^{th}$ and $75^{th}$ percentiles of $0.0121$ $\mathrm{cm \, s^{-1}}$ and $0.0303 \, \mathrm{cm \, s^{-1}}$ respectively). The calculated ozone deposition velocity varies from $0.009 \, \mathrm{cm \, s^{-1}}$ in polar waters to $0.040 \, \mathrm{cm \, s^{-1}}$ at the tropics. This improves comparisons to observations. The variability is driven mainly by the temperature-dependant rate constant for the reaction between iodide and ozone, the temperature dependence of the solubility and variations in the ocean iodide concentration. The calculated annual deposition flux of ozone to the ocean is reduced from $222 \, \mathrm{Tg \, yr^{-1}}$ to $112 \, \mathrm{Tg \, yr^{-1}}$, and overall deposition of ozone to all surface types reduces from $862 \, \mathrm{Tg \, y^{-1}}$ to $758 \, \mathrm{Tg \, y^{-1}}$. Tropospheric ozone burdens and global mean OH increase from $324 \, \mathrm{Tg}$ to $328 \, \mathrm{Tg}$, and from $1.17 \times 10^6 \, \mathrm{molec \, cm^{-3}}$ to $1.18 \times 10^6 \, \mathrm{molec \, cm^{-3}}$, respectively. 34% of surface grid boxes experience a 10% or greater increase in ozone concentration. Comparisons between observations of surface ozone and the model are improved with the new parameterization notably around the Southern Ocean. Process level representation of oceanic deposition of ozone thus appears essential for representing the concentration of surface ozone over the planet.

## 1 Introduction

Tropospheric ozone is an important secondary pollutant. Globally it causes one million premature deaths a year (Malley et al., 2017), degrades ecosystems (The Royal Society, 2008) and is a greenhouse gas (IPCC Stocker et al., 2013). Transport from the stratosphere and in-situ chemical production are balanced by chemical destruction and dry deposition to the surface. Total dry deposition of ozone is thought to amount to $\sim 978 \, \mathrm{Tg \, y^{-1}}$ (Hardacre et al., 2015) compared to $\sim 500 \, \mathrm{Tg \, y^{-1}}$ transported from the stratosphere, $\sim 5000 \, \mathrm{Tg \, y^{-1}}$ for chemical production, and $\sim 4500 \, \mathrm{Tg \, y^{-1}}$ for chemical loss (Young et al., 2018). Whilst





dry deposition velocity to the ocean is thought to be slow ($\sim 0.05$ cm s$^{-1}$) compared to vegetation ($\sim 0.1$ cm s$^{-1}$), the larger area of the ocean compared to the land results in ozone deposition to the ocean representing approximately one third of the total deposition (Ganzeveld et al., 2009).

The ultimate sink of ozone to the ocean is due to chemical reactions. The reaction of ozone with iodide ($[I^-]$)in the surface layer

of the ocean via the simplified reaction R1 (Garland and Curtis, 1981; Sakamoto et al., 2009; Hayase et al., 2010; Carpenter et al., 2013) is believed to be the dominant mechanism (Garland et al., 1980). The transport of ozone within the ocean surface also plays an important role in this process, a simplified version of the relevant processes is shown in Fig. 1.

$$O_3 + I^- + H^+ \rightarrow HOI + O_2 \tag{R1}$$

In addition, dissolved organic carbon (DOC) has been shown to react with dissolved ozone and have an enhancing effect

on ozone deposition similar to that of iodide (Martino et al., 2012; Shaw and Carpenter, 2013), but is less well understood. Dimethyl sulfide (DMS) and bromide have also been shown to enhance ozone deposition velocity but by small amounts (Sarwar et al., 2016).

The net flux of a gas to a surface $F$ is calculated as the atmospheric concentration at the ocean surface $C$ multiplied by the deposition velocity, $v_d$, shown in equation 1.

$$F = v_d C \tag{1}$$

The deposition velocity ($v_d$) in many models is calculated using the resistance-in-series scheme (Wesely and Hicks, 1977) shown in equation 2. This describes the different limiting factors of the deposition: transport to the surface through turbulent transport ($r_a$); transport through the quasilaminar sub-layer, which is the air directly in contact with a surface ($r_b$); and the chemical or biological destruction of the molecule at the surface (the ocean in this case) ($r_c$).

$$v_d = \frac{1}{r_a + r_b + r_c} \tag{2}$$

The relative importance of the different resistances is dependent primarily on the gas being considered. Gases that are highly soluble (such as sulfur dioxide) giving them a small $r_c$, so their limiting factors are the atmospheric resistances ($r_a$ and $r_b$). Less soluble gases such as ozone are limited by the chemical loss at the surface ($r_c$). Wesely (1989) gives a value of $r_c = 2000$ s m$^{-1}$ for ozone in all water types, and this is used in most atmospheric chemistry models (Hardacre et al., 2015; Luhar et al.,

2017, 2018). This chemical loss of ozone, is the limiting factor for ozone deposition (95% of the sum of the resistances is the value of $r_c$ (Chang et al., 2004)) and so yields an almost constant (0.05 cm s$^{-1}$) overall deposition velocity, with only small variation due to meteorological variation in $r_a$ and $r_b$. However, observations of ozone deposition show significant variability. From the observations collated by Ganzeveld et al. (2009), fresh water deposition velocities range from 0.01 to 0.1 cm s$^{-1}$, with ocean observations ranging from 0.01 to 0.15 cm s$^{-1}$. The higher values of ocean observations are likely influenced by

coastal effects such as those described by Bariteau et al. (2010), with the open ocean observations being substantially lower (0.009 - 0.065 cm s$^{-1}$) (Helmig et al., 2012).

Given this observed variability, the fixed $r_c$ approach appears overly simple. Based on Fairall et al. (2007) and Luhar et al.





(2017), Luhar et al. (2018) formulated a new scheme for calculating $r_c$ which explicitly takes into account the simultaneous effects of chemical reactions in the ocean with iodide and the physical processes of molecular diffusion and turbulent transfer in the ocean surface. This considers three oceanic layers (Fig. 1); a very shallow "surface reaction-diffusion" layer, that represents the region of the ocean through which the $O_3$ can diffuse from the ocean before it reacts in the ocean, which lies above a thicker

turbulent layer which is mixed by wind-stress driven turbulence, which in turn, lies above the the 'bulk' ocean. The loss of $O_3$ is determined by the chemical reactivity within the reaction-diffusion layer, which is supplied by $I^-$ from below. The resulting scheme derived by Luhar et al. (2018), is based on solving the fundamental equation for the conservation of mass of a reacting and diffusing substance in water (Fairall et al., 2007), yields equation 3

$$r_c = \frac{1}{\alpha\sqrt{aD}}\left[\frac{\Psi K_1(\xi_\delta)sinh(\lambda) + K_0(\xi_\delta)cosh(\lambda)}{\Psi K_1(\xi_\delta)cosh(\lambda) + K_0(\xi_\delta)sinh(\lambda)}\right] \tag{3}$$

where $\alpha$ is the dimensionless solubility, $a$ the chemical reactivity of $O_3$ with sea-surface iodide (the product of $[I^-]$) and the second order rate-coefficient ($k$)), $D$ the diffusivity of $O_3$ in water, $\Psi$ is defined in equation 5 where $u_w^*$ is the water-side friction velocity, $\delta_m$ is the thickness of the reaction-diffusion layer of the sea-surface microlayer, $\kappa$ the von Kármán constant ($\approx 0.4$), $\xi_\delta$ defined in equation 4, $\lambda$ defined in equation 6 and $K_0$, $K_1$ are modeified Bessel funcitons of the second kind order zero and one respectively. where,

$$\xi_\delta = \left[\frac{4a}{\kappa u_w^*}\left(\delta_m + \frac{D}{\kappa u_w^*}\right)\right]^{\frac{1}{2}} \tag{4}$$

$$\Psi = \left[1 + \left(\frac{\kappa u_w^* \delta_m}{D}\right)\right]^{\frac{1}{2}} \tag{5}$$

$$\lambda = \delta_m\sqrt{\frac{a}{D}} \tag{6}$$

In this paper we include this description of ozone deposition to the ocean into the GEOS-Chem model and explore the impact on the composition of the troposphere. In Section 2 we describe the GEOS-Chem model and the implementation of the new scheme. In Section 3 we describe the impact of the new scheme on the deposition velocities of ozone to the ocean in the model and assess them against observations of deposition velocities. The impacts of the new deposition scheme on the composition of the troposphere are described in Section 4 together with comparison to observations of surface ozone . Finally we draw some

conclusions in Section 5.

## 2 Modeling

We use here Version 12.1.1 of the 3-D chemical transport model GEOS-Chem Classic (Bey et al., 2001) (www.geos-chem.org) driven by assimilated meteorology from the NASA Global Modeling and Assimilation Office. GEOS-Chem includes HOx-NOx-VOC-ozone-halogen-aerosol tropospheric chemistry with the halogen (chlorine, bromine and iodine) chemistry being the





most recent addition as described by Sherwen et al. (2016b). In this work we use global simulations run at a spatial resolution of 2°x2.5° with meteorological data from MERRA-2 (Gelaro et al., 2017). We run simulations for 2006-2008, 2013 and 2014 so that field observations are compared with the appropriate meteorology. Analysis of the sensitivity of the ozone deposition velocity to its controlling factors uses model runs for 2014. For the analysis of the impact on atmospheric composition, a one

year 'spin-up' was used to allow the tropospheric composition to reach equilibrium before the subsequent analysis year.

As with many other atmospheric chemistry and transport models, the dry deposition in GEOS-Chem uses a resistance-in-series scheme based on that of Wesely (1989). The details of this implementation are described by Wang et al. (1998). For terrestrial land types, the dry deposition in GEOS-Chem is generally consistent with observations (Silva and Heald, 2018).

We follow the Luhar et al. (2018) methodology, and as shown in Equation 3, this requires the calculation of $\alpha, D, k, [I^-]$ and

$\delta_m$. Where these require the sea surface temperature ($T$) we use the skin temperature from the MERRA-2 meteorological fields.

We use the dimensionless solubility of ozone in water $\alpha$ from Morris (1988)

$$\alpha = 10^{-0.25-0.013(T-273.16)} \tag{7}$$

the diffusivity D ($\mathrm{m^2 s^{-1}}$) from Johnson and Davis (1996).

$$D = 1.1 \times 10^{-6} exp\left(\frac{-1896}{T}\right) \tag{8}$$

the temperature dependent $k$ ($\mathrm{M^{-1} s^{-1}}$) for the aqueous phase reactions between ozone and iodide from Magi et al. (1997)

$$k = exp\left(\frac{-8772.2}{T} + 51.5\right) \tag{9}$$

the reaction-diffusion sublayer thickness (m) is defined as

$$\delta_m = \sqrt{\frac{D}{a}} \tag{10}$$

and the global ocean iodide concentration distribution $[I^-]$ (M) is taken from the most recent global climatology (Sherwen et al., 2019).

The waterside friction velocity $u_w^*$ ($\mathrm{m\,s^-1}$) can be calculated from the MERRA-2 atmospheric friction velocity $u^*$ using equation 11 where $\rho_a$ and $\rho_w$ are the density of the atmosphere and seawater respectively. This assumes that drivers of atmospheric stress result in an equivalent oceanic stress (Fairall et al., 2007).

$$u_w^* = \sqrt{\frac{\rho_a}{\rho_w}} u^* \approx 0.0345 u^* \tag{11}$$

Three significant differences exist in our choice of parameters compared to the work of Luhar et al. (2018). Firstly, we use the Sherwen et al. (2019) ocean iodide distributions, whereas they use MacDonald et al. (2014). Sherwen et al. (2019) is based on a recent collation of sea surface iodide observations (Chance et al., 2019) which are interpolated using a machine learning approach. MacDonald et al. (2014) is based on a more restrictive observational dataset and uses a simple temperature based





parameterization. Sherwen et al. (2019) calculates a global average sea-surface [$I^-$] of $105.8 \pm 45.6$ nM which is a significant increase from the global mean of $58.9 \pm 34.9$ nM found by MacDonald et al. (2014). Secondly, we include a variable thickness for the reaction-diffusion sublayer (Equation 10). Luhar et al. (2018) explore various options for this parameter and decide upon a fixed value of $\delta_m$ ($3.0 \times 10^{-6}$ m) as this gave the best fit of $v_d$ to observations from Helmig et al. (2012). We decide

to use the variable definition in our work as this is more physically based and produces comparable results in our simulations. However, it should be noted that using this definition of $\delta_m$ results in terms cancelling in equation 6 such that $\lambda = 1$. This thus simplifies equation 3 somewhat as $sinh(1) \approx 1.175$ and $cosh(1) \approx 1.543$. Some of the implications for different choices for $\delta_m$ are explored in Luhar et al. (2018). Finally, we differentiate between salt and fresh water, using a salinity map from the World Ocean Atlas 2013 (Zweng et al., 2013). The new ozone deposition scheme is only applied to ocean water. Anywhere

with water and a salinity below 20 PSU or no salinity value (fresh water) is assigned a constant $r_c = 2000$ s m$^{-1}$.

## 3 Impact of new parameterization on deposition

### 3.1 Change in global distribution of deposition velocities

Figure 2 shows the annual average global distribution of oceanic ozone deposition velocity for both the standard model and the updated surface resistance scheme, along with the percentage difference between the two. Table 1 gives a statistical description

of global ozone dry deposition in the model. The near uniform value of $v_d$ with the standard uniform surface resistance can be observed in Fig. 2 (top). The small variability in deposition velocity seen is driven by differences in the meteorology. This contrasts with the variability calculated with the new scheme (middle). The two schemes also differ in the magnitude of the deposition velocities. The largest change occurs in the coolest waters towards the poles, with the Southern Ocean having a reduction of over 90% compared to the standard scheme, whereas the tropics can have as little as a 10% reduction. The

distribution of $v_d$ is similar to that shown in Luhar et al. (2018), despite our use of the variable thickness for the reaction-diffusion sublayer and the use of the Sherwen et al. (2016a) iodide. On an area-weighted basis, the deposition of ozone to the ocean surface is reduced from 0.0464 cm s$^{-1}$ ($25^{th}$ and $75^{th}$ percentiles of 0.0461 cm s$^{-1}$ and 0.0471 cm s$^{-1}$ respectively), to 0.0231 cm s$^{-1}$ ($25^{th}$ and $75^{th}$ percentiles of 0.0121 cm s$^{-1}$ and 0.0303 cm s$^{-1}$ respectively). This amounts to a halving of the mean ocean deposition velocity. The reduction of deposition velocity to the ocean results in a reduction of 17% in the

global average deposition velocity (table 2). The total annual loss of tropospheric ozone to dry deposition decreases by 104 Tg y$^{-1}$ to 758 Tg y$^{-1}$, substantially lower than the average of $978 \pm 127$ Tg y$^{-1}$ from the multi-model comparison found by Hardacre et al. (2015) but comparable to the value obtained by Luhar et al. (2018) of 722 Tg y$^{-1}$. The seasonal changes in ozone oceanic deposition velocities from the new annual mean are shown in Fig. 3. This shows the response of the ozone deposition velocity to changes in sea-surface temperature with the highest value in the summer for each hemisphere and the

lowest values occurring in the winter. In the extra-tropical oceans, deposition velocities are predicted to vary by roughly 50% between summer and winter. Deposition velocities in the tropics remain relatively constant over the year.





## 3.2 Comparison to observations

Here we evaluate the modelled deposition velocities against the open ocean measurements from Helmig et al. (2012) who measured ozone fluxes to the ocean surface using eddy covariance. These measurements are from a series of five cruises between 2006 to 2008 that took place in the Gulf of Mexico, eastern Pacific Ocean, western Atlantic Ocean and Southern

Ocean (Fig. 4). These cruises were made it waters of significantly different sea surface temperature (SST)and show a trend between deposition velocity and the SST. The old scheme (grey line) overestimates the rate of dry deposition substantially and fails to capture any of the temperature dependencies seen in the observations. The new scheme (black line) is a significant improvement, agreeing more with the magnitude and the temperature dependence of the observations. It should be noted that there are significant uncertainties in the measured deposition velocities at low values (Helmig et al., 2012). Combining all the

measurements made by Helmig et al. (2012) and comparing to the model predictions for deposition velocity, the root mean square error for the model agreement was reduced from $0.04 \ \mathrm{cm\,s^{-1}}$ using the default scheme to $0.01 \ \mathrm{cm\,s^{-1}}$ using the new scheme. Whilst the overall agreement of the model with the observations has been improved, the model still fails to capture all of the variability of the deposition velocity measurements. This may be an issues with the resolution of the model ( $2°x2.5°$) which may fail to capture local conditions. Uncertainties in sea-surface iodide concentration or the lack of other sea-surface

reactions (reaction between ozone and DOC) may also contribute.

## 3.3 Sensitivity of new scheme

We explore here the sensitivity of the new scheme to our choice of parameterization for $u_w^*$, $I^-$, $k$, $D$ and $\alpha$. Five model simulations were each run for a year with only one of the parameters allowed to vary. When constrained, the value of each parameter was set to a representative value of the global average ($\alpha, D, k$ calculated with an SST of 289 K, sea-surface iodide concen-

tration of 106 nM, and $u_w^*$ of $0.01 \ \mathrm{m\,s^{-1}}$). A sixth model simulation was run with all $r_c$ parameters kept constant at these representative values. The resulting dependence of deposition velocity for each simulation is shown in Fig. 5 as a function of sea surface temperature. If all of the terms needed to calculate $r_c$ are kept constant (top left) the oceanic deposition velocity does not vary with temperature. Similarly, if only the water side friction velocity is allowed to vary, no dependence on temperature is seen. Surprisingly the temperature dependence of the iodide concentration is not large, reflecting its square root

dependence in the calculation of $r_c$. The two most important factors for giving the observed temperature dependence are $k$ and $\alpha$. Of these two terms, the dependence on rate coefficient carries the most uncertainty.

Magi et al. (1997) is the only temperature dependent rate constant in the literature. Other studies are at single temperatures and show differences (Luhar et al., 2018). We explore the impact of these differences by running a number of simulations with different values of the rate constants (Fig. 6). We use the single temperature rate constants given by Garland et al. (1980)

($2.0 \times 10^9 \ \mathrm{M^{-1}s^{-1}}$ at 298K), Liu et al. (2001) ($1.2 \times 10^9 \ \mathrm{M^{-1}s^{-1}}$ at 298K) and Hu et al. (1995) ($4.0 \times 10^9 \ \mathrm{M^{-1}s^{-1}}$ at 277K).





We also use the upper (equation 12) and lower (equation 13) estimates of Magi et al. (1997) (based on the reported error of the series of measurements).

$$k = exp\left(\frac{-9261.6}{T} + 53.6\right) \tag{12}$$

$$k = exp\left(\frac{-8796.2}{T} + 50.8\right) \tag{13}$$

Figure 6 shows that the uncertainties in $k$ can substantially impact the modeled deposition velocity, with the difference between a temperature invariant and temperature dependent $k$ most notable. The temperature independent rate constants don't correctly simulate the observed temperature variability in deposition velocity. The higher estimate from Magi et al. (1997) over estimates the deposition velocity in warm waters, with the lower estimate underestimating in cold waters. As discussed in section 1 iodide is the dominant but not only removal mechanism for ozone at the ocean surface. Given the upper and mid value of the Magi et al. (1997) rate constants there does not appear to be much potential role for other oceanic components to play an important role. On the other hand if the lower values of the Magi et al. (1997) rate constant were correct, this would allow for inclusion of additional reactions (such as that of ozone with dissolved organic carbon) in the model parameterization without overestimating deposition velocities.

## 4 Atmospheric impact

### 4.1 Global impacts

The net decrease in deposition of ozone to the surface results in an increase in both surface and column ozone mixing ratio (Fig. 7). The greatest increase in ozone concentration occurs in the boundary layer with the magnitude of the change decreasing with altitude through the troposphere. The largest increases in ozone mixing ratio is above the oceans, most notably the extra-tropics, with increases becoming negligible over land. Surface grid boxes that experience a 10% increase or greater in ozone mixing ratio represent 34% of the total surface grid box count. Table 2 gives diagnostics on the oxidative capacity of the troposphere for both the old and new schemes. The increase in ozone mixing ratio shown in Fig. 7 equates to an increase in the tropospheric ozone burden of 4 $\mathrm{Tg\,y^{-1}}$ (1.2%). This effects the global chemical production and loss of $O_3$, however these changes are globally minimal at -0.6% and 1.2%, respectively.

Another consequence of the increased ozone mixing ratio is a small increase in global mean OH concentration of 0.9% (table 2), resulting in a decrease in the tropospheric methane lifetime from 8.3 years to 8.2 years.

Seasonal variations are also observed in the changes in surface ozone mixing ratio due to the new scheme (Fig. 8). The largest increase is observed over the oceans during the winter of each hemisphere due to both the lower deposition velocity that occurs in colder waters and due to the dry deposition playing a larger role in the ozone budget when photolysis is at a seasonal low.





## 4.2 Regional impacts

To assess the predictions of surface ozone mixing ratios in the model, comparisons were made with surface ozone measurements from a number of World Meteorological Organization (WMO) Global Atmosphere Watch (GAW; http://www.wmo.int/pages/prog/arep/gaw/gaw_home_en.html, accessed through EBAS http://ebas.nilu.no/, the database infrastructure operated by
NILU – Norwegian Institute for Air Research) sites around the world (Fig. 9, shown south to north).

The largest area of change in surface ozone in the model is in the Southern Ocean. GAW sites in this region (Cape Grim, Ushuaia and Neumayer) show increases in ozone prediction during their winter/spring with the increase most notable in the Antarctic site of Neumayer. Previous work in GEOS-Chem by Schmidt et al. (2016) and Sherwen et al. (2016a) as well as inter-model comparison with ozonesonde observations by Young et al. (2013) show a low bias of GEOS-Chem and other models
in the Southern Ocean and Antarctic region. The increased surface ozone mixing ratio brings the model predictions closer to the observations in the Southern Ocean region (Fig. 9), as well as the reductions in root mean square error (RMSE), a measure of disagreement between the model and observations, (table 3) which is reduced by an average of 44% across these three locations. Whilst there are considerable improvements in the Antarctic location of Neumayer, surface ozone demonstrate a 'lag' in responding to Antarctic spring/summer. The model also fails to capture the spring time halogen induced ozone depletion
events that are observed at Neumayer.

A comparison to a clean tropical location is made using the GAW site in Cape Verde. Tropical waters are where there has been the least change in ozone deposition velocity, as well as the least increase in ozone mixing ratio both annually and seasonally. Whilst there is a slight increase in predicted ozone compared to the observations at Cape Verde both the model using the old and new schemes for ozone deposition are within the error of the observations, and there is a small reduction in RMSE.
Mace Head, Ireland offers an evaluation of model performance in a mid-latitude inflow region, the inflow of air from the North Atlantic at this site is the dominant component into Europe. Comparing the increase to the observations at Mace Head the improvement is notable with the models error reduced by approximately 30%.

The most northerly of the GAW sites in this comparison is the Villum research station in Greenland. There is a minimal increase in predicted surface ozone (∼1 ppbv) at this site and the resulting RMSE (table 3) shows for Villum an increases of 0.3
ppbv with the new parameterization. The observations at Villum also show spring time ozone depletion events and, as with Neumayer, the model fails to capture this.

Overall, the majority of GAW sites show an improved comparisons with observations due to the implementation of the new $r_c$ scheme and supporting that this change is an improvement to the model.

## 5 Conclusions

We have implemented a new scheme for the deposition of ozone to the ocean into the GEOS-Chem chemistry transport model based on the work of Luhar et al. (2018). This considers the physical and chemical controls of ozone loss in the sea surface. In contrast to Luhar et al. (2018), our work has used a variable surface micro-layer depth and the higher ocean iodide concentrations from Sherwen et al. (2019). The new scheme results in a halving of the global mean ozone deposition



velocity to the ocean, leading to a small increase in the global tropospheric ozone burden and some regional increases in ozone mixing ratios of up to 30% in the high latitude boundary layer, notably around the Southern Ocean. The new scheme improves comparisons between the model and observations in oceanic regions. The increase in tropospheric ozone concentration also has a minor effect on the global mean OH and $CH_4$ lifetimes.

5    The new parameterization improves comparisons between the model and observed oceanic dry deposition velocities. However, no account has been made of potential additional processes such as the reaction of $O_3$ with DOC, DMS and bromide at the ocean surface. Uncertainties in the rate constant for the reaction between $I^-$ and $O_3$ could allow room for such additional reactions to play a role. Reduced uncertainty in the temperature dependent rate constant for this reaction would be useful. In addition it seems likely that the interaction between DOC and ozone would be complex. It seems likely that some compounds

10    will act as deposition enhancers, whilst others may act as inhibitors (Martino et al., 2012; Shaw and Carpenter, 2013). Further lab, field and modeling studies will be required to better constrain this.



*Code availability.* GEOS-Chem version 12.1.1 used in this project DOI: 10.5281/zenodo.2249246 . This code will be implemented into forthcoming update of the GEOS-Chem available from https://github.com/geoschem/geos-chem/tree/master Code is also available on request.

*Author contributions.* R.J.P performed model development, ran the simulations and analysed the output. M.J.E., L.J.C. and R.J.P developed

5  the project. T.S assisted in model development. D.H. provided ozone deposition data. Paper was written by R.J.P with contributions from all co-authors.

*Competing interests.* The authors declare that they have no conflict of interest.

*Acknowledgements.* We thank WMO GAW and the individual sites that make up this network, for the availability of the surface ozone data. This project was undertaken on the Viking Cluster, which is a high performance compute facility provided by the University of York. We are

10  grateful for computational support from the University of York High Performance Computing service, Viking and the Research Computing team.

R.J.P. thanks NERC SPHERES DTP (NE/L002574/1) for funding his PhD studentship.

L.J.C., M.J.E., T.S. thank funding from Natural Environment Research Council (NERC) through the grant "Iodide in the ocean: distribution and impact on iodine flux and ozone loss" (NE/N009983/1)

15  L.J.C. acknowledges funding from the European Research Council (ERC) under the European Union's horizon 2020 programme (Grant agreement No. 833290)





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

©c Author(s) 2019. CC BY 4.0 License.





Helmig, D., Lang, E. K., Bariteau, L., Boylan, P., Fairall, C. W., Ganzeveld, L., Hare, J. E., Hueber, J., and Pallandt, M.: Atmosphere-ocean ozone fluxes during the TexAQS 2006, STRATUS 2006, GOMECC 2007, GasEx 2008, and AMMA 2008 cruises, J GEOPHYS RES-ATMOS, 117, https://doi.org/10.1029/2011JD015955, 2012.

Hu, J. H., Shi, Q., Davidovits, P., Worsnop, D. R., Zahniser, M. S., and Kolb, C. E.: Reactive Uptake of Cl2(g) and Br2(g) by Aqueous Surfaces as a Function of Br- and I- Ion Concentration: The Effect of Chemical Reaction at the Interface, J. Phys. Chem., 99, 8768–8776, https://doi.org/10.1021/j100021a050, 1995.

IPCC Stocker, T., Qin, D., Plattner, G., Tignor, M., Allen, S., Boschung, J., Nauels, A., Xia, Y., Bex, V., and Midgley, P.: Climate Change 2013: The Physical Science Basis. Contribution of Working Group I to the Fifth Assessment Report of the Intergovernmental Panel on Climate Change, 2013.

Johnson, P. N. and Davis, R. A.: Diffusivity of Ozone in Water, J CHEM ENG DATA, 41, 1485–1487, https://doi.org/10.1021/je9602125, 1996.

Liu, Q., Schurter, L. M., Muller, C. E., Aloisio, S., Francisco, J. S., and Margerum, D. W.: Kinetics and Mechanisms of Aqueous Ozone Reactions with Bromide, Sulfite, Hydrogen Sulfite, Iodide, and Nitrite Ions, INORG CHEM, 40, 4436–4442, https://doi.org/10.1021/ic000919j, 2001.

Luhar, A. K., Galbally, I. E., Woodhouse, M. T., and Thatcher, M.: An improved parameterisation of ozone dry deposition to the ocean and its impact in a global climate-chemistry model, ATMOS CHEM PHYS, 17, 3749–3767, https://doi.org/10.5194/acp-17-3749-2017, 2017.

Luhar, A. K., Woodhouse, M. T., and Galbally, I. E.: A revised global ozone dry deposition estimate based on a new two-layer parameterisation for air–sea exchange and the multi-year MACC composition reanalysis, ATMOS CHEM PHYS, 18, 4329–4348, https://doi.org/10.5194/acp-18-4329-2018, https://www.atmos-chem-phys.net/18/4329/2018/, 2018.

MacDonald, S. M., Gómez Martín, J. C., Chance, R., Warriner, S., Saiz-Lopez, A., Carpenter, L. J., and Plane, J. M. C.: A laboratory characterisation of inorganic iodine emissions from the sea surface: dependence on oceanic variables and parameterisation for global modelling, ATMOS CHEM PHYS, 14, 5841–5852, https://doi.org/10.5194/acp-14-5841-2014, https://www.atmos-chem-phys.net/14/5841/2014/, 2014.

Magi, L., Schweitzer, F., Pallares, C., Cherif, S., Mirabel, P., and George, C.: Investigation of the Uptake Rate of Ozone and Methyl Hydroperoxide by Water Surfaces, J PHYS CHEM A, 101, 4943–4949, https://doi.org/10.1021/jp970646m, 1997.

Malley, C. S., Henze, D. K., Kuylenstierna, J. C., Vallack, H. W., Davila, Y., Anenberg, S. C., Turner, M. C., and Ashmore, M. R.: Updated Global Estimates of Respiratory Mortality in Adults $\geq$ 30 Years of Age Attributable to Long-Term Ozone Exposure, ENVIRON HEALTH PERSP, 125, 087 021, https://doi.org/10.1289/EHP1390, 2017.

Martino, M., Lézé, B., Baker, A., and Liss, P.: Chemical controls on ozone deposition to water, Geophys. Res. Lett., 39, L05 809, https://doi.org/10.1029/2011GL050282., 2012.

Morris, J.: The aqueous solubility of ozone - A review, Ozone news, 1, 14–16, 1988.

Sakamoto, Y., Yabushita, A., Kawasaki, M., and Enami, S.: Direct Emission of I2 Molecule and IO Radical from the Heterogeneous Reactions of Gaseous Ozone with Aqueous Potassium Iodide Solution, J PHYS CHEM A, 113, 7707–7713, https://doi.org/10.1021/jp903486u, 2009.

Sarwar, G., Kang, D., Foley, K., Schwede, D., and Gantt, B.: Technical note: Examining ozone deposition over seawater, ATMOS ENVIRON, 141, 255–262, https://doi.org/10.1016/j.atmosenv.2016.06.072, 2016.

Schmidt, J. A., Jacob, D. J., Horowitz, H. M., Hu, L., Sherwen, T., Evans, M. J., Liang, Q., Suleiman, R. M., Oram, D. E., Le Breton, M., Percival, C. J., Wang, S., Dix, B., and Volkamer, R.: Modeling the observed tropospheric BrO background: Importance of multiphase chemistry





and implications for ozone, OH, and mercury, J GEOPHYS RES-ATMOS, 121, 11,819–11,835, https://doi.org/10.1002/2015JD024229, 2016.

Shaw, M. D. and Carpenter, L. J.: Modification of Ozone Deposition and I₂ Emissions at the Air-Aqueous Interface by Dissolved Organic Carbon of Marine Origin, ENVIRON SCI TECHNOL, 47, 10 947–10 954, https://doi.org/10.1021/es4011459, 2013.

Sherwen, T., Evans, M. J., Carpenter, L. J., Andrews, S. J., Lidster, R., Dix, B., Koenig, T. K., Sinreich, R., Ortega, I., Volkamer, R., Saiz-Lopez, A., Prados-Roman, C., Mahajan, A. S., and Ordóñez, C.: Iodine's impact on tropospheric oxidants: a global model study in GEOS-Chem, ATMOS CHEM PHYS, 16, 1161–1186, https://doi.org/10.5194/acp-16-1161-2016, https://www.atmos-chem-phys.net/16/1161/2016/, 2016a.

Sherwen, T., Schmidt, J. A., Evans, M. J., Carpenter, L. J., Großmann, K., Eastham, S. D., Jacob, D. J., Dix, B., Koenig, T. K., Sinreich, R.,
Ortega, I., Volkamer, R., Saiz-Lopez, A., Prados-Roman, C., Mahajan, A. S., and Ordóñez, C.: Global impacts of tropospheric halogens (Cl, Br, I) on oxidants and composition in GEOS-Chem, ATMOS CHEM PHYS, 16, 12 239–12 271, https://doi.org/10.5194/acp-16-12239-2016, https://www.atmos-chem-phys.net/16/12239/2016/, 2016b.

Sherwen, T., Chance, R. J., Tinel, L., Ellis, D., Evans, M. J., and Carpenter, L. J.: A machine-learning-based global sea-surface iodide distribution, EARTH SYST SCI DATA, 11, 1239–1262, https://doi.org/10.5194/essd-11-1239-2019, https://www.earth-syst-sci-data.net/
15    11/1239/2019/, 2019.

Silva, S. J. and Heald, C. L.: Investigating Dry Deposition of Ozone to Vegetation, J GEOPHYS RES-ATMOS, 123, 559–573, https://doi.org/10.1002/2017JD027278, 2018.

The Royal Society: Ground-level ozone in the 21st century: future trends, impacts and policy implications, Policy Document, 2008.

Wang, Y., Jacob, D. J., and Logan, J. A.: Global simulation of tropospheric O3-NO x -hydrocarbon chemistry: 1. Model formulation, J
GEOPHYS RES-ATMOS, 103, 10 713–10 725, https://doi.org/10.1029/98JD00158, 1998.

Wesely, M.: Parameterization of surface resistances to gaseous dry deposition in regional-scale numerical models, ATMOS ENVIRON, 23, 1293–1304, 1989.

Wesely, M. and Hicks, B.: Some Factors that Affect the Deposition Rates of Sulfur Dioxide and Similar Gases on Vegetation, Journal of the Air Pollution Control Association, 27, 1110–1116, https://doi.org/10.1080/00022470.1977.10470534, 1977.

Young, P., Naik, V., Fiore, A., Gaudel, A., Guo, J., Lin, M., Neu, J., Parrish, D., Rieder, H., Schnell, J., Tilmes, S., Wild, O., Zhang, L., Ziemke, J., Brandt, J., Delcloo, A., Doherty, R., Geels, C., Hegglin, M., Hu, L., Im, U., Kumar, R., Luhar, A., Murray, L., Plummer, D., Rodriguez, J., Saiz-Lopez, A., Schultz, M., Woodhouse, M., and Zeng, G.: Tropospheric Ozone Assessment Report: Assessment of global-scale model performance for global and regional ozone distributions, variability, and trends., Elem Sci Anth, 6(1), 10, https://doi.org/http://doi.org/10.1525/elementa.265, 2018.

Young, P. J., Archibald, A. T., Bowman, K. W., Lamarque, J.-F., Naik, V., Stevenson, D. S., Tilmes, S., Voulgarakis, A., Wild, O., Bergmann, D., Cameron-Smith, P., Cionni, I., Collins, W. J., Dalsøren, S. B., Doherty, R. M., Eyring, V., Faluvegi, G., Horowitz, L. W., Josse, B., Lee, Y. H., MacKenzie, I. A., Nagashima, T., Plummer, D. A., Righi, M., Rumbold, S. T., Skeie, R. B., Shindell, D. T., Strode, S. A., Sudo, K., Szopa, S., and Zeng, G.: Pre-industrial to end 21st century projections of tropospheric ozone from the Atmospheric Chemistry and Climate Model Intercomparison Project (ACCMIP), ATMOS CHEM PHYS, 13, 2063–2090, https://doi.org/10.5194/acp-13-2063-2013,
https://www.atmos-chem-phys.net/13/2063/2013/, 2013.

Zweng, M., Reagan, J., Antonov, J., Locarnini, R., Mishonov, A., Boyer, T., Garcia, H., Baranova, O., Johnson, D., D.Seidov, and Biddle, M.: World Ocean Atlas 2013 Volume 2: Salinity, NOAA Atlas NESDIS 74, p. 39, 2013.





**Table 1.** Area-weighted annual average deposition velocity and deposition flux by land type for ozone in GEOS-Chem using the default (constant) and new (variable) scheme for calculating $r_c$. The $25^{th}$ and $75^{th}$ percentiles are the subscripts and superscripts respectively for each land types deposition velocity

| Land type | Constant $r_c$ | | Variable $r_c$ | |
| --- | --- | --- | --- | --- |
| | $O_3$ $v_d$ [cm s$^{-1}$] | $O_3$ deposition flux [Tg yr$^{-1}$] | $O_3$ $v_d$ [cm s$^{-1}$] | $O_3$ deposition flux [Tg yr$^{-1}$] |
| Land | $0.2370_{0.1486}^{0.2612}$ | 383 | $0.2370_{0.1486}^{0.2612}$ | 386 |
| Ocean | $0.0464_{0.0461}^{0.0471}$ | 222 | $0.0231_{0.0121}^{0.0303}$ | 122 |
| Mixed* | $0.1501_{0.0489}^{0.1785}$ | 255 | $0.1426_{0.0332}^{0.1755}$ | 248 |
| Ice | $0.0098_{0.0094}^{0.0100}$ | 2 | $0.0098_{0.0094}^{0.0100}$ | 2 |
| All | $0.0937_{0.0319}^{0.0582}$ | 862 | $0.0781_{0.0124}^{0.0460}$ | 758 |

*Where mixed is defined as any grid box containing less than 100% water and less than 100% land





**Table 2.** Summary of change to atmospheric oxidative capacity for GEOS-Chem using default (constant) scheme for calculating $r_c$ and the new scheme (variable)

| | Constant | Variable |
|---|---|---|
| Troposphere $O_3$ burden [Tg] | 324 | 328 |
| Net chemical $O_3$ rate [$Tg\,y^{-1}$]* | 450 | 363 |
| $O_X$ production rate [$Tg\,y^{-1}$]* | 5048 | 5017 |
| $O_X$ loss rate [$Tg\,y^{-1}$] * | 4598 | 4654 |
| $O_3$ loss to deposition [$Tg\,y^{-1}$] | 862 | 758 |
| Stratospheric $O_3$ flux [$Tg\,y^{-1}$] | 412 | 395 |
| Global annual mean OH [$10^6\,molec\,cm^{-3}$] | 1.17 | 1.18 |
| Global $CH_4$ lifetime [years] | 8.3 | 8.2 |

*with $O_X$ defined as $O_3 + NO_2 + NO_3 + HNO_4 + HNO_3 + N_2O_5 + BrO + HOBr + BrNO_2 + BrNO_3 + IO + HOI + IONO + IONO_2 + OIO + I_2O_2 + I_2O_3 + I_2O_4 ClO + HOCl + ClNO_2 + ClNO_3 + Cl_2O_2 + OClO + PAN$ (peroxyacetylnitrate) + PMN (Peroxymethacryloylnitrate) + PPN (Peroxypropionylnitrate) + MPN (Methyl peroxy nitrate) + ETHLN (Ethanal nitrate) + $R_4N_2$ ($\geq C_4$ alkylnitrates) + R4N1 (RO2 from $R_4N_2$) + Isoprene Nitrate (ISN1, ISOPNB, ISOPND, ISNP) + Peroxy radical from isoprene (ISNOOA, ISNOOB, ISNOHOO) + MACRN (Methacrolein nitrate) + MVKN (nitrate from methly vinyl keytone) + PROPNN (propanone nitrate) + $O_2NOCH_2C(OO)(CH_3)CH{=}CH_2$ (INO2) + $O_2NOCH_2C(OOH)(CH_3)CH{=}CH_2$ (INPN) + $HOCH_2C(ONO_2)(CH_3)CHO$ (MAN2) + PRN1 (RO2 from propene + NO3) + PRPN (Peroxide from PRN1) + MACRNO2 (result of $HOCH_2C(ONO_2)(CH_3)CHO + OH$). For further details on this tagging see the GEOS-Chem wiki http://wiki.seas.harvard.edu/geos-chem/index.php/FlexChem



**Table 3.** Root mean square error (RMSE) of the model with the default (constant) scheme for $r_c$ and the new scheme (variable) when compared to the observations at GAW sites calculated from monthly mean values of observations and model predictions.

| GAW site | Constant RMSE [ppbv] | Variable RMSE [ppbv] |
|----------|----------------------|----------------------|
| Villum | 4.2 | 4.5 |
| Mace Head | 5.0 | 3.4 |
| Cape Verde | 2.6 | 2.0 |
| Cape Grim | 3.5 | 1.5 |
| Ushuaia | 2.7 | 2.0 |
| Neumayer | 5.6 | 2.8 |





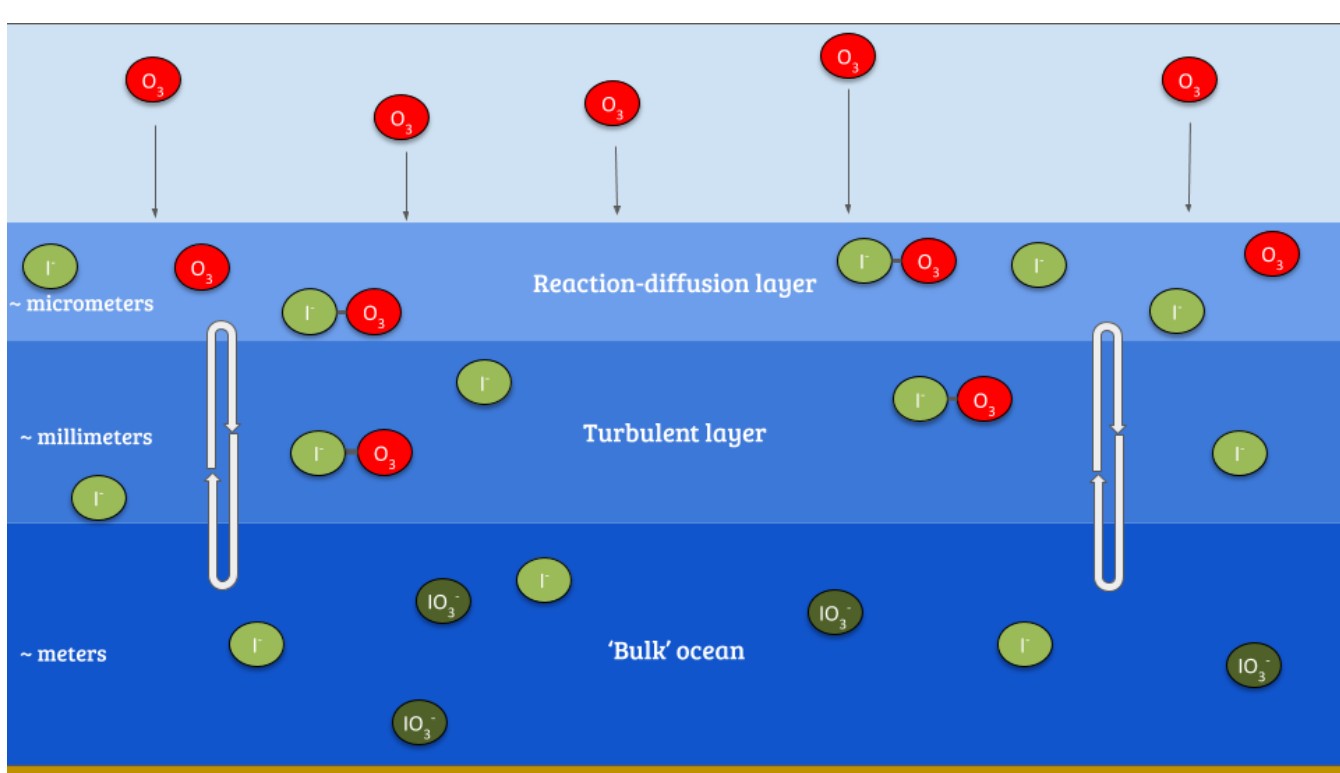

**Figure 1.** Illustration of the reaction of ozone with $I^-$ in the sea-surface also demonstrating a simplified version of the surface structure where the reaction occurs







**Figure 2.** Annual average ozone deposition velocities for 2014 as calculated by GEOS-Chem using the default deposition scheme (top), the new parameterization (middle) and the percentage change between the two schemes (bottom). A 2°x2.5° land mask has been applied to the deposition velocities to show only the deposition velocity to the ocean.

**Figure 3.** Percentage change from the annual mean deposition velocity for December, January, Febuary (DJF) March, April, May (MAM) June, July August (JJA) and September, October, November (SON) for the new parameterization (shown in figure 2) demonstrating the deposition velocity responding to changes in sea-surface temperature and ocean $I^-$ concentration with the lowest values of deposition velocity seen in the winter of each hemisphere. Land and ice grid boxes have been masked out.

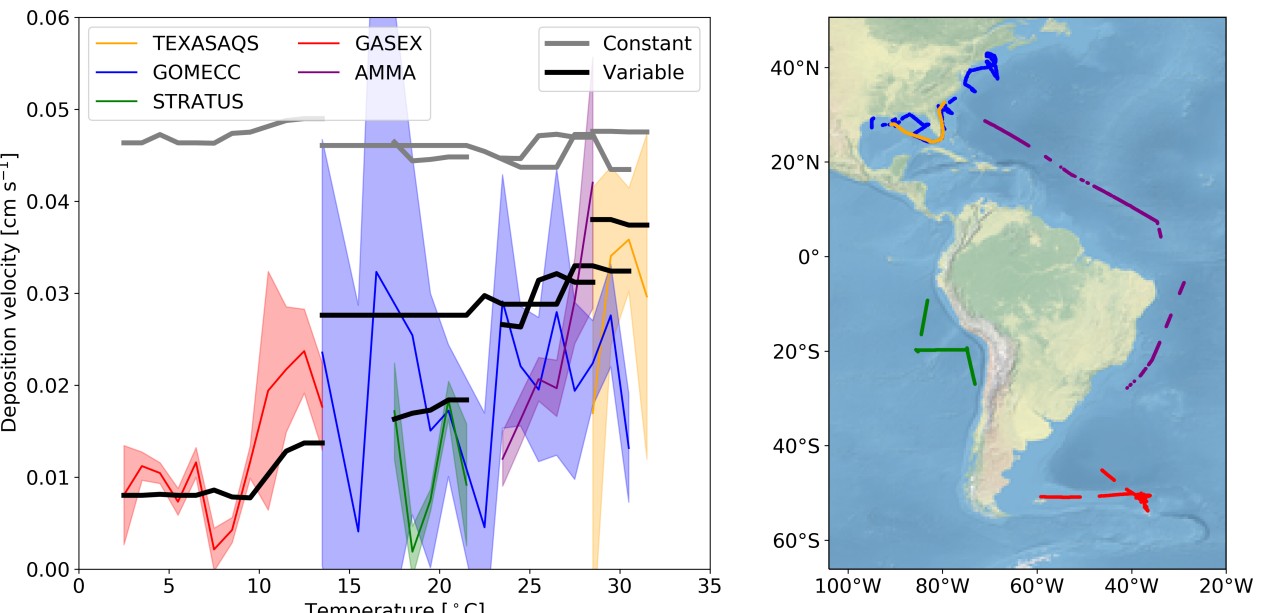

**Figure 4.** (left) The deposition velocities predicted by the model using the default (Constant) value for $r_c$ and the new (Variable) parameterization of $r_c$ compared against the 5 open ocean cruise data-sets of ozone deposition by Helmig et al. (2012). The solid lines representing the median of the deposition velocity for a one degree temperature window, with the shaded region representing the $25^{th}$ to $75^{th}$ percentiles. (right) The locations along the cruise tracks where the ozone deposition has been compared.





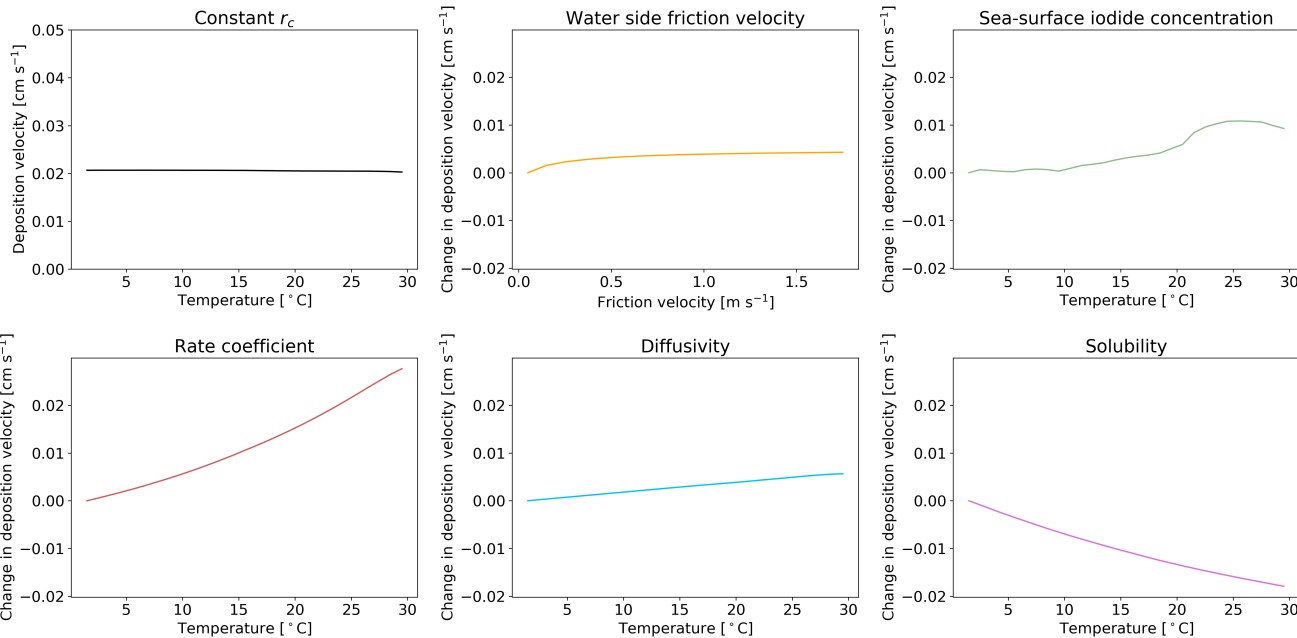

**Figure 5.** The response of deposition velocity to the variation of only a single parameter with other parameters set to global average values. Each function is produced from global values averaged into 1K temperature bins.



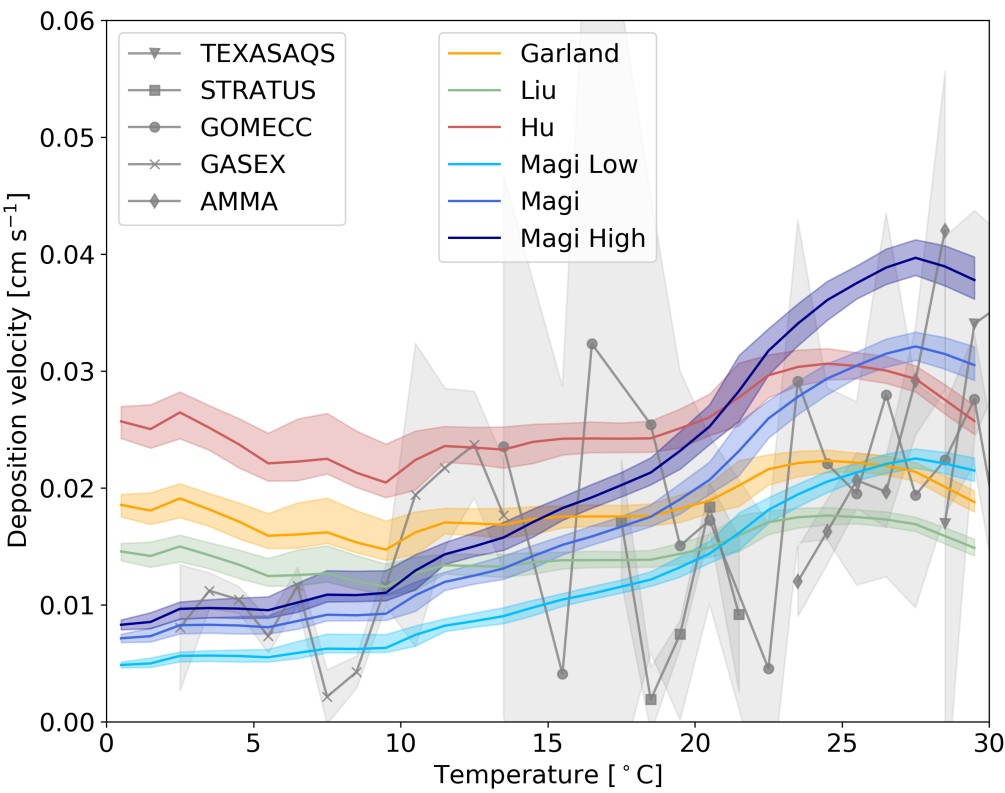

**Figure 6.** The response of deposition velocity to different different laboratory measurements of $k$. Three are constant with respect to temperature (Garland et al., 1980; Liu et al., 2001; Hu et al., 1995) and the temperature dependent parameterization of Magi et al. (1997) with two additional cases of $k$ based on the error range of the Magi et al. (1997) measurements (shown in equation 13 and 12). Each function is produced from global values averaged into 1K temperature bins.

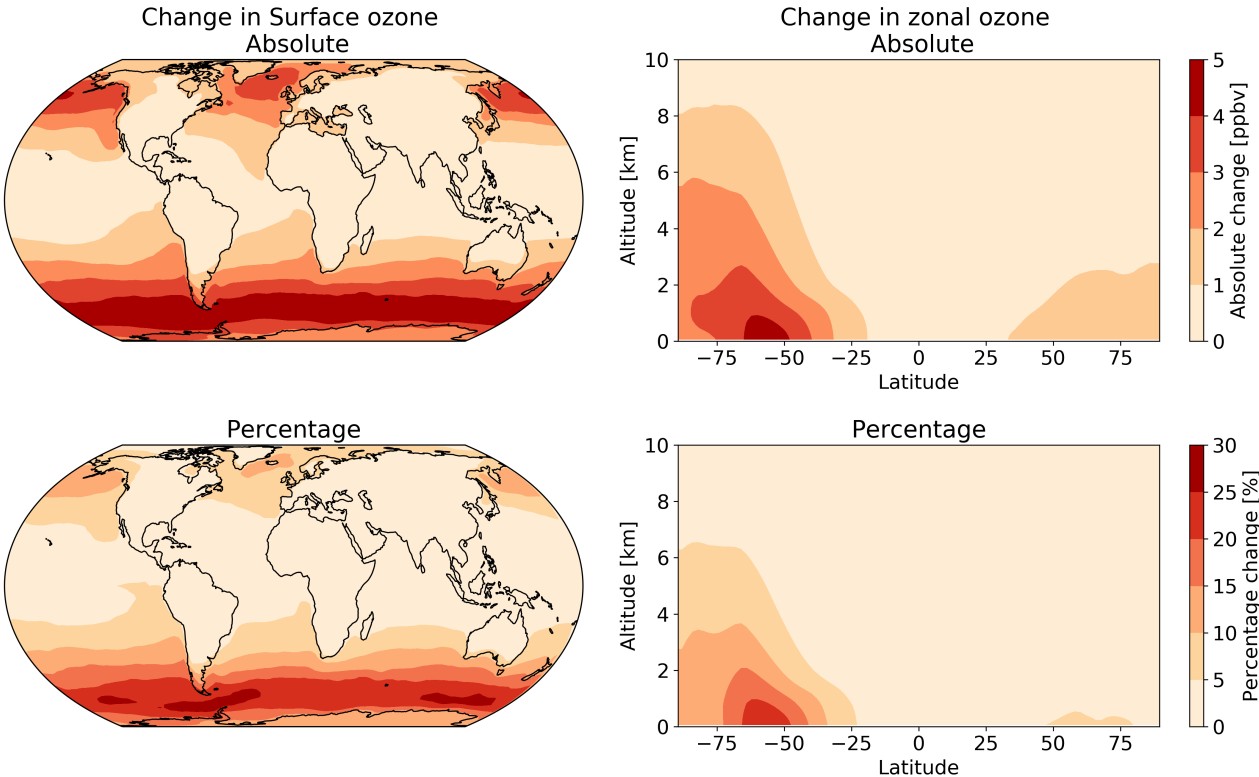

**Figure 7.** The annual absolute (first row) and percentage (second row) change in surface and column ozone mixing ratios between the model using the default (constant) and new (variable) parameterization for $r_c$. The largest changes occur in the surface levels of the model, especially in higher latitudes with the Southern Ocean boundary layer representing the area experiencing the most annual average change between the two model runs.

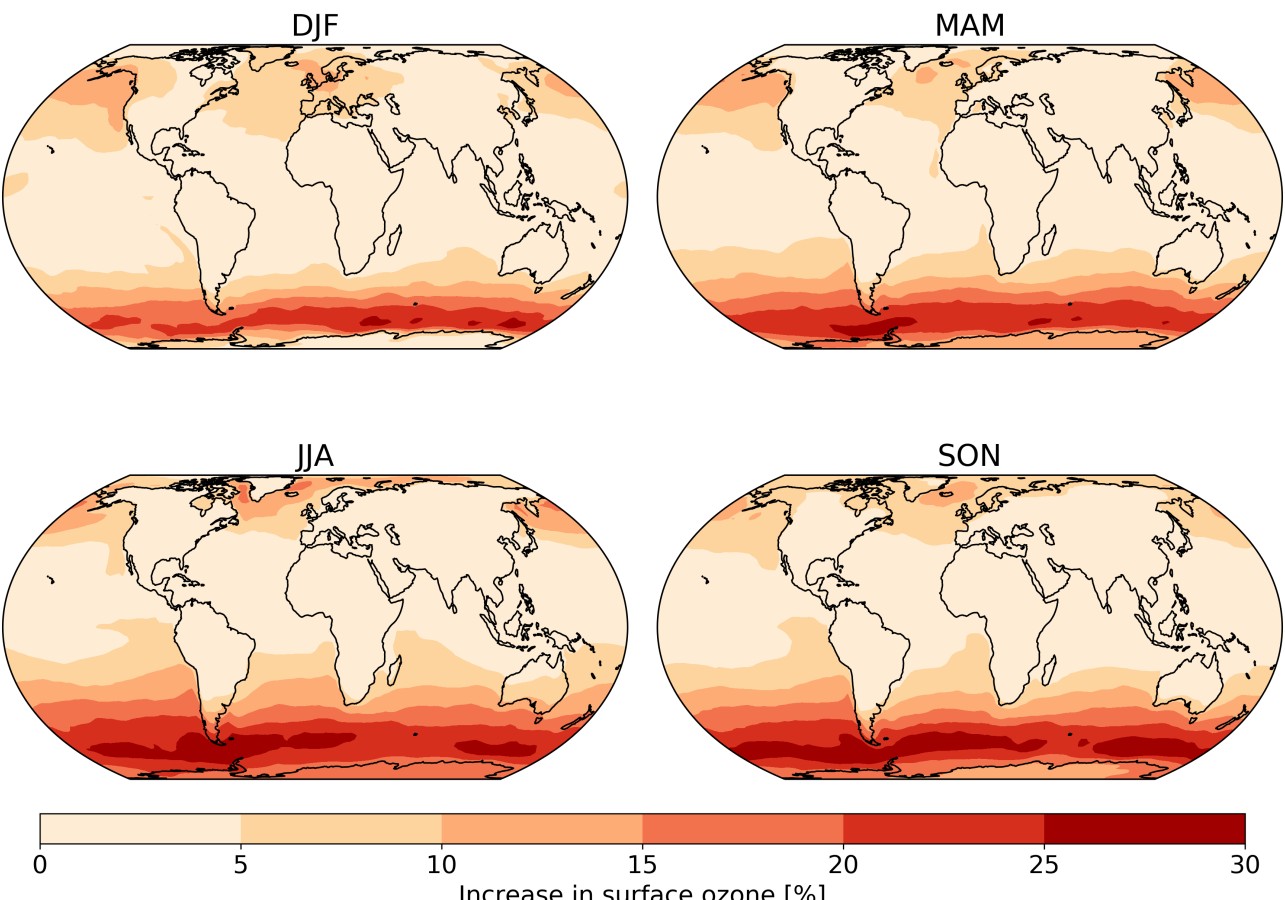

**Figure 8.** The absolute seasonal surface ozone mixing ratio change between the model runs using the default (constant) and new (variable) parameterization for $r_c$.



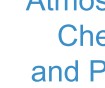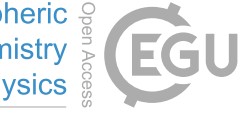

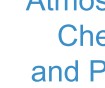

**Figure 9.** Predictions and observations of monthly average surface ozone mixing ratio from the model using the default (Constant) and new (Variable) parameterization for $r_c$ for six GAW stations (with the latitude and longitude for each station at the bottom right).