# Peer review of "Influences of oceanic ozone deposition on tropospheric photochemistry"

_Atmospheric Chemistry and Physics, 2019_

## Referee Comment (RC1) · Anonymous Referee #1 · 9 Dec 2019

This paper implements a new process-based scheme for ozone dry deposition to the ocean based on the work of Luhar et al. (2017, 2018) into the widely-used GEOS-Chem model. The impact of this implementation on parameters pertaining to tropospheric photochemistry, such as tropospheric ozone and its budget, methane life time and OH burden, is reported. The sensitivity of the new scheme to the O3–I reaction rate constant is also highlighted. It is shown that on average there is a significant improvement in the modelled ozone compared to measurements when the new oceanic deposition scheme is used in place of the default Wesely scheme.

Overall, the paper is interesting, well written, and is suitable for publication in ACP subject to the following, largely minor, comments:

1. In Section 3.2 and Fig. 4, the modelled deposition velocities are compared with the

data from Helmig et al. It is not clear how they are matched in space and time. Please provide some details. What was the temporal resolution of the model output for vd that was used in the comparison with the data?

Following on from the above point, if an appropriate temporal resolution (e.g. hourly) model output for vd is available, perhaps it will also be useful to plot vd vs. wind speed from the model and compare with the corresponding data plot in Helmig et al. (their Fig. 5, solid lines).

2. Table 1. 25th and 75th percentiles are given. It is not clear as to over what kind of sample size/properties (e.g. based on hourly or monthly modelled values?) these statistics are calculated?

3. Page 4, last para: I think another point of difference between the present model and that used by Luhar et al. (2018) is that the former includes halogen chemistry. Perhaps this should be mentioned.

4. Abstract, line 13: 112 Tg yr-1. In Table 1, it is given as 122 Tg yr-1.

5. Page 2, line 19: Generally, in addition to chemical or biological destruction, rc can also include physical loss at the surface.

6. Page 3, line 13: 'second kind order' to 'second kind with order'.

7. Page 3, line 27: '3-D chemical' to '3-D global chemical'.

8. Eq. 3: The functions sinh and cosh are usually not italicised.

9. Page 4, line 10: Give the unit of SST (K or deg. C) used in the parametrisation equations.

10. Page 5, line 25: I think 'table 1' should be 'table 2'.

11. Page 5, line 25 and elsewhere: Luhar et al.'s global deposition value is presented. It would be useful to also present the uncertainty they calculated, i.e. 722 $\pm$ 87.3 Tg

yr-1.

12. Page 6, line 5: 'it' to 'in'.

13. Page 6, line 17: Put spaces around 'and' and no italics.

14. Page 7, lines 18-19: You could also highlight that the largest increases are in the extra tropics and they are more pronounced in the Southern Hemisphere.

15. Table 2: What year(s) do these values correspond to?

16. Table 3. What year(s) do these values correspond to?

17. Figure 3. Please put year (2014?) in the caption. The same for Figs. 7–9.

---

## Referee Comment (RC2) · Anonymous Referee #2 · 17 Dec 2019

Pound et al. present research implementing an updated oceanic ozone deposition scheme into a chemical transport model. The deposition of ozone and other compounds to both land and water is an important uncertainty in global models of atmospheric chemistry. The manuscript addresses a key uncertainty in modern models, is well written, and likely suitable for publication in ACP once the following comments are addressed.

**Major Comments**
**Resolution Dependence**
The simulations in this work are completed at 2˚x2.5˚ globally. Dry deposition is highly dependent on environmental conditions that vary in their distribution at higher spatial resolutions. It's not immediately clear from the text that the performance assessment in this work would be consistent at higher model resolutions. Do the authors expect their implementation to be a similar improvement at all relevant spatial resolutions, or are these results unique to 2˚x2.5˚?

**Computational Expense**
A major advantage of the simplified fixed surface resistance is the associated light computational burden in calculating deposition velocities. It would be useful if the authors could comment on the additional computational expense (if any) of their improved simulated deposition velocities.

**Additional Species**
The parameterization presented here is likely to be relevant and useful to the simulation of species other than just ozone. It would be valuable to the broader community if the authors could comment on what would be necessary to extend this analysis to other chemical species, and potentially what the impact on those species would be.

**Minor Comments**
Eq 1: In the atmospheric science literature, dry deposition velocities are typically written with respect to the atmosphere (e.g. $F = -V_d * C$). The sign in this equation is unclear with respect to the reference frame of the deposition.

P6 L26: The labels k and $\alpha$ are inconsistent between the text and Figure 5, which uses full name descriptions. This adds confusion for the reader.

Figure 5: Why are all panels a function of temperature except for "Water side friction velocity"? Is the Water side friction velocity also binned by temperature?

Figure 6 & Figure 9: What do the shaded regions in the figure represent?

**Technical Edits**

P2 L21: "Gases that are highly soluble giving them a small $r_c$". This sentence is confusing as written.

P3 L10: "(the product of …" the parenthesis in this section appear to be off.

P6 L17: "*Dand$\alpha$*" authors likely meant "*D* and $\alpha$"

---

## Author Comment (AC1) · 20 Feb 2020

**Authors response to Anonymous Referee #1 comments**

1. *In Section 3.2 and Fig. 4, the modelled deposition velocities are compared with the data from Helmig et al. It is not clear how they are matched in space and time. Please provide some details. What was the temporal resolution of the model output for vd that was used in the comparison with the data? Following on from the above point, if an appropriate temporal resolution (e.g. hourly) model output for vd is available, perhaps it will also be useful to plot vd vs. wind speed from the model and compare with the corresponding data plot in Helmig et al. (their Fig. 5, solid lines).*

   **Response**: The comparison between cruise and model observations is done on a daily average for the grid box and model data from grid boxes selected based on the latitude and longitude coordinates of the ship during that 24 hr period. Due to this temporal resolution the wind speed comparison would not be appropriate.

   **"***The comparisons between observations and model were made using daily average values with model output selected from grid boxes the ship track passed through in that 24 hour period.***"**

2. *Table 1. 25th and 75th percentiles are given. It is not clear as to over what kind of sample size/properties (e.g. based on hourly or monthly modelled values?) these statistics are calculated?*

   **Response**: The table description has been updated to give more information about how these statistics were calculated.

   **"**...*The average deposition velocities, 25th and 75th percentiles were calculated from monthly average model values for grid boxes containing 100% of the land type specified unless otherwise stated.***"**

3. *Page 4, last para: I think another point of difference between the present model and that used by Luhar et al. (2018) is that the former includes halogen chemistry. Perhaps this should be mentioned.*

   **Response**: The authors agree with the referee's comment and included the further difference between this work and that of Luhar et al. (2018) by drawing attention to the difference in chemistry schemes, specifically that GEOS-Chem includes halogen chemistry.

   **"***One further difference between this work and that of Luhar et al. (2018) is in the global chemistry transport model and its chemistry scheme, GEOS-Chem includes halogen chemistry which has a notable effect on tropospheric ozone (Sherwen et al., 2016b)***"**

**4.** *Abstract, line 13: 112 Tg yr-1. In Table 1, it is given as 122 Tg yr-1.*

**Response:** This was a typo and the value in the abstract has been updated. This sentence now reads "*The calculated annual deposition flux of ozone to the ocean is reduced from 222 Tg yr−1 to 122 Tg yr−1 ...*"

**5.** *Page 2, line 19: Generally, in addition to chemical or biological destruction, rc can also include physical loss at the surface.*

**Response**: The sentence has been updated as the referee suggests to correctly include the possibility of physical loss as a contributing factor to rc. The sentence now reads "*...and the physical, chemical or biological loss of the molecule at the surface...*"

**6.** *Page 3, line 13: 'second kind order' to 'second kind with order'.*

**Response**: Sentence updated as per reviewers comment to now read "*... modified Bessel functions of the second kind with order zero and one respectively.*"

**7.** *Page 3, line 27: '3-D chemical' to '3-D global chemical'.*

**Response**: Sentence updated as per reviewers comments to now correctly read "*... the 3-D global chemical transport model GEOS-Chem Classic...*"

**8.** *Eq. 3: The functions sinh and cosh are usually not italicised.*

**Response**: As per the reviewers comment the formula has been updated so that the sinh and cosh functions are no longer italicised.

$$r_c = \frac{1}{\alpha\sqrt{aD}}\left[\frac{\Psi K_1(\xi_\delta)\sinh(\lambda) + K_0(\xi_\delta)\cosh(\lambda)}{\Psi K_1(\xi_\delta)\cosh(\lambda) + K_0(\xi_\delta)\sinh(\lambda)}\right]$$

**9.** *Page 4, line 10: Give the unit of SST (K or deg. C) used in the parameterization equations.*

**Response**: Specified units of kelvin for T, with sentence now reading "*...sea surface temperature (K), T...*"

**10.** *Page 5, line 25: I think 'table 1' should be 'table 2'.*

**Response**: Updated reference to correctly point to table 1

**11.** *Page 5, line 25 and elsewhere: Luhar et al.'s global deposition value is presented. It would be useful to also present the uncertainty they calculated, i.e. 722 ± 87.3 Tg yr-1*

**Response**: The uncertainty in the value of total global deposition of ozone from Luhar et al. (2018) has now been included where this figure is quoted.

*12. Page 6, line 5: 'it' to 'in'.*

**Response**: Corrected typo by changing 'it' to 'in' so that sentence now reads **"***These cruises were made in waters of significantly different...***"**

*13. Page 6, line 17: Put spaces around 'and' and no italics.*

**Response**: Corrected typo, removing 'and' from math mode such that it is now correctly spaced and no longer italicised. Sentence now reads "...,k,D and α."

*14. Page 7, lines 18-19: You could also highlight that the largest increases are in the extra tropics and they are more pronounced in the Southern Hemisphere.*

**Response**: Based on the referee's comment, the wording of this sentence has been changed to specifically highlight for the reader that the Southern Hemisphere extra tropics see the largest increase in surface ozone concentration. This sentence now reads

**"***...most notably the extra-tropics with the Southern Hemisphere extra-tropics being the area of greatest increase. The increase in surface ozone concentration 20 becomes negligible over land.***"**

*15. Table 2: What year(s) do these values correspond to?*

**Response**: The table caption has been updated to specify that the values in the table are calculated from a model run of the year 2014. The caption now reads

**"***Area-weighted annual average deposition velocity and deposition flux for 2014…***"**

*16. Table 3. What year(s) do these values correspond to?*

**Response**: The table caption has been updated to specify that the values in the table are calculated from a model run of the year 2014. The caption now reads

**"***Summary of change to atmospheric oxidative capacity for GEOS-Chem using default (constant) scheme for calculating rc and the new scheme (variable) for 2014***"**

*17. Figure 3. Please put year (2014?) in the caption. The same for Figs. 7–9.*

**Response**: The captions of Figures 3,7,8 and 9 have been updated to state that they are showing model data from 2014.

---

## Author Comment (AC2) · 20 Feb 2020

**Authors response to Anonymous Referee #2 comments**

1. *Resolution Dependence: The simulations in this work are completed at 2°x2.5° globally. Dry deposition is highly dependent on environmental conditions that vary in their distribution at higher spatial resolutions. It's not immediately clear from the text that the performance assessment in this work would be consistent at higher model resolutions. Do the authors expect their implementation to be a similar improvement at all relevant spatial resolutions, or are these results unique to 2°x2.5°?*

**Response**: We do not believe there is any resolution dependence in this calculation of deposition velocity and results at higher resolutions would still be comparable to those we achieved at 2°x2.5°. The following addition has been made to the paper to address this

**"*Whilst 2°x2.5° is a relatively coarse model resolution, we don't believe that there is any significant sub-grid scale correlation between tropospheric ozone concentration and sea-surface I⁻ concentration therefore this should not result in a resolution dependence*"**

2. *Computational Expense: A major advantage of the simplified fixed surface resistance is the associated light computational burden in calculating deposition velocities. It would be useful if the authors could comment on the additional computational expense (if any) of their improved simulated deposition velocities.*

**Response**: We agree that the simplified method for calculating rc would have a light computational burden. The nature of dry deposition calculation this would be negligible with respect to the rest of the model and have made an addition to the paper to state this. We do not have appropriate diagnostics to provide a quantitative statement on the additional burden.

**"*Any additional computational expense of implementing this improved $r_c$ calculation will be small as the deposition velocity calculation remains a two dimensional problem, unlike the chemistry or transport calculations which are three dimensional problems.*"**

3. *Additional Species: The parameterization presented here is likely to be relevant and useful to the simulation of species other than just ozone. It would be valuable to the broader community if the authors could comment on what would be necessary to extend this analysis to other chemical species, and potentially what the impact on those species would be.*

**Response**: We agree with the reviewers comment that mention of how to apply this deposition scheme to other species and what would be required to do so would be a useful addition to the discussion made in this paper. The following addition has been made to the paper.

**"***It would be possible to apply this method of calculating $r_c$ to other chemical species. If the appropriate sink processes were understood, chemical kinetics available, and concentrations of reactant species known. For this to be useful, the species would need to have a high dependence on $r_c$ (rather than the physical resistances), but also for dry deposition to form a substantial part of the species budget. It is not clear whether any species, other than $O_3$, would meet these requirements.***"**

4.  *Eq 1: In the atmospheric science literature, dry deposition velocities are typically written with respect to the atmosphere (e.g. F = -V d * C). The sign in this equation is unclear with respect to the reference frame of the deposition.*

**Response**: The in text description of equation 1 does describe the direction of the flux as towards the surface but as commented by the referee equation 1 is inconsistent with this.
The formula for equation 1 has been updated to correctly reflect this.

$$F = -u_d C$$

5.  *P6 L26: The labels k and a are inconsistent between the text and Figure 5, which uses full name descriptions. This adds confusion for the reader.*

**Response**: The figure caption for figure 5 has been updated to give the labels for each of the full name descriptions and wording of the caption updated to address this comment and comment 6.

6.  *Figure 5: Why are all panels a function of temperature except for "Water side friction velocity"? Is the Water side friction velocity also binned by temperature?*

**Response**: As the referee's comment states the figure caption was misleading and this has been updated to correctly state that all functions are binned by temperature apart from water side friction velocity which is binned by friction velocity. The caption now reads

**"***Figure 5. The response of deposition velocity to the variation of only a single parameter with other parameters set to global average values. Sea-surface iodide concentration [I −], rate coefficient k, diffusivity D and solubility α are produced from global values averaged into 1 K temperature bins. Water side friction velocity u * w is averaged into 0.1 m s−1 friction velocity bins.***"**

7.  *Figure 6 & Figure 9: What do the shaded regions in the figure represent?*

**Response**: For both figure 6 and figure 9 the captions have been updated to state that this shaded region represents the 25th to 75th percentile range.

8.  *P2 L21: "Gases that are highly soluble giving them a small r c ". This sentence is confusing as written.*

**Response**: We agree that the wording of this sentence could be confusing so this sentence has been reworded and now reads

"*Gases that are highly soluble (such as sulfur dioxide) will have a small rc, so their limiting factors are the atmospheric resistances (ra and rb)*"

9. *P3 L10: "(the product of ..." the parenthesis in this section appear to be off.*

**Response**: The surplus parenthesis in this sentence has been removed such that the sentence now reads as intended

"*...(the product of [I−] and the second order rate-coefficient, k)...*"

10. *P6 L17: "Dandα" authors likely meant "D and α"*

**Response**: Corrected typo, removing 'and' from math mode such that it is now correctly spaced and no longer italicised. Sentence now reads "*...,k,D and α.*"